# Etiologic Factors of Temporomandibular Disorders: A Systematic Review of Literature Containing Diagnostic Criteria for Temporomandibular Disorders (DC/TMD) and Research Diagnostic Criteria for Temporomandibular Disorders (RDC/TMD) from 2018 to 2022

**DOI:** 10.3390/healthcare12050575

**Published:** 2024-02-29

**Authors:** Joanna Warzocha, Joanna Gadomska-Krasny, Joanna Mrowiec

**Affiliations:** 1Faculty of Medicine, Lazarski University, Świeradowska 43, 02-662 Warszawa, Poland; 2Ordo-Dent. Gabinet Stomatologiczny, Edmunda Bałuki 22, 70-406 Szczecin, Poland; drgadomska@wp.pl; 3SCS Astermed-Centrum Ortodontyczno-Implantologiczne, Świętego Bonifacego 92, 02-940 Warszawa, Poland

**Keywords:** temporomandibular disorders, TMD, diagnostic criteria for temporomandibular disorders, research diagnostic criteria for temporomandibular disorders, etiology, biopsychosocial model

## Abstract

This study aims to conduct a systematic analysis of literature published between 1 January 2018 and 1 September 2022, exploring factors influencing the progression or development of temporomandibular disorders (TMD), diagnosed using the Diagnostic Criteria for Temporomandibular Disorders (DC/TMD) or Research Diagnostic Criteria for Temporomandibular Disorders (RDC/TMD). Three electronic databases were reviewed to identify papers that examined TMD factors using DC/TMD or RDC/TMD. Inclusion criteria encompassed original research published in English between 1 January 2018 and 1 October 2022, online, and complete DC/TMD or RDC/TMD studies on human participants aged 18 or older. Two authors independently assessed the risk of bias using The Joanna Briggs Institute (JBI) Analytical cross-sectional studies’ Critical Appraisal Tool. Of 1478 articles, 11 were included. The studies revealed strong associations between TMD and factors such as female, poor sleep quality, depression, oral parafunction, anxiety, somatization, and anatomical features. However, variables such as education, living conditions, socioeconomic status, marital status, chronic pain, and stress did not exhibit statistically significant correlations. Based on the obtained data, it can be concluded that the causes of TMD are largely related to psychological factors, which supports the biopsychosocial theory of the disorder.

## 1. Introduction

Temporomandibular disorder (TMD) includes a range of musculoskeletal disorders affecting the masticatory muscles, the temporomandibular joints (TMJ), and surrounding tissue structures [1]. The pathophysiological causes of this condition may arise from alterations in the structure and function of the TMJ joints themselves, or the surrounding muscles and/or other tissues.

Rather than a single cause, TMD arises from a multifactorial interplay of various factors, including biochemical changes, such as structural abnormalities, muscle dysfunction, trauma, genetic mutations, hormonal changes, systematic diseases, and other factors. According to the literature, whiplash injury is considered to be a specific type of trauma that has been associated with the development of this disorder. The prevalence of TMD among patients with whiplash injury has been reported to range from 14% to 37.5% [2]. Hypertension and insulin resistance are recognized as significant disease factors that are becoming more prevalent in the population and have an impact on the development of TMD. The literature reports on the effect of raised blood pressure on the impairment of central pain regulatory systems and its potential contribution to painful TMD. Patients with temporomandibular disorder (TMD) exhibit heightened sensitivity to aversive stimuli, implying that the presence of painful TMD may arise, to some extent, from dysfunction in central pain modulation mechanisms influenced by baseline arterial blood pressure [3]. Helena Martynowicz et al. [4] suggest that hypertension, higher BMI, lower values of mean SpO2, and higher percentages of SpO2 < 90% constitute independent risk factors for increased bruxism episode index. It has been suggested that autoimmune and inflammatory disorders could potentially play a role in the development of TMD. According to a study conducted by Ji Rak Kim et al. [5], a small percentage of subjects (15%) exhibited ANA/RF positivity. Furthermore, Shinya Kototaki et al. [6] reported a high prevalence of severe TMD in patients with SAPHO syndrome. According to Alina Grozdinska et al. [7], there appears to be a statistical relationship between TMD and Hashimoto’s thyroiditis (HT). In their study, muscle pain and stiffness were present in 86.5% of HT patients, and disc displacement in 63.4%. Additionally, Ehlers–Danlos syndrome is another systemic disorder that has been linked to TMD. According to Karen Bech et al. [8], there is a significant correlation between hypermobile Ehlers–Danlos syndrome and symptoms and signs of TMD, as well as osseous changes in the TMJs. It has also been noted that genetics and TMD have been associated with congenital coagulation disorders. According to the clinical study conducted by Selda Yenel et al. [9], it was found that patients with inherited coagulation disorders, especially hemophilia, may have a higher likelihood of developing temporomandibular disorders (TMD) compared to healthy individuals.

Symptoms of TMD can include discomfort and pain in the orofacial region, limited TMJ mobility, difficulty with speech and chewing, stiffness, tinnitus, and clicking or skipping sounds when chewing, opening, or closing the mouth. It is essential to objectively diagnose TMD to manage them effectively. The persistent and progressive manifestation of symptoms deteriorates the quality of life and psychological well-being, which potentially impacts existing psychiatric ailments like depression, chronic stress, and anxiety. Wiackiewicz et al. suggest that Polish patients with TMD have heightened levels of anxiety, depression, perceived stress, and pain intensity, recommending screening assessments utilizing Patient Health Questionnaire-9, Perceived Stress Scale-10, and Generalized Anxiety Disorder-7 [10]. Seweryn et al. [11] proved that a large number of TMD patients experiencing poor sleep quality and the associated reduced life satisfaction; these parameters should be considered as influential factors that modify the management of patients with TMD. Research suggests that psychological disorders have an impact on TMD development [12]. Considering the multifactorial etiology of TMD, their development is explained by the biopsychosocial model [13,14].

TMD is an umbrella term, encompassing multifactorial and heterogeneous disorders that may occur in different genders and ages. According to epidemiological data, this problem affects from 5–12% [15] of the population, to 21.5–50.5% [16], and is the second most common musculoskeletal dysfunction, after chronic lower back pain [17]. The systematic review by Valesan et al. [1] showed that the overall prevalence of temporomandibular joint disorders was approximately 31% for adults/elderly and 11% for children/adolescents. Minervini et al. [18] found that TMD prevalence in children and adolescents varies between 20% and 60% and females had a higher prevalence of TMD compared to males. A 2020 study found that the frequency of TMD among the Polish urban adult population was 55.9% [19]. Patients with TMD require multi-specialist management, often extending beyond dentists’ expertise.

To our knowledge, previous literature reviews on the etiology of TMD have not used a single consistent tool. In contrast, this study utilized both the DC/TMD and RDC/TMD questionnaires to standardize the diagnosis of TMD and objectively identify its causal factors. Additionally, a large study group was used as an inclusion and exclusion criterion, further enhancing the objectivity of the results. All of these factors have contributed to a paper that discusses various aspects of the etiology of TMD. This is relevant for both clinicians and from a scientific perspective. Recognizing the identified links between psychological factors and TMD, it is imperative to enhance the identification of psychological conditions in at-risk individuals. Effective early intervention programs necessitate development, incorporating a comprehensive management approach with collaboration among diverse professionals.

The objective of this study is to provide a systematic analysis of the literature published from 1 January 2018 to 1 September 2022 concerning the factors that affect the development and progression of TMD in patients who received a diagnosis by using the Diagnostic Criteria for Temporomandibular Disorders (DC/TMD) or Research Diagnostic Criteria for Temporomandibular Disorders (RDC/TMD) protocol.

## 2. Materials and Methods

This review was conducted following the Preferred Reporting Items for Systematic Reviews and Meta-Analysis (PRISMA) guidelines and was registered in PROSPERO (ID: CRD42024497070, date: 15 January 2024).

### 2.1. Eligibility Criteria and Information Sources

All selected papers met the criteria. Three online databases were searched: Medline Complete, PubMed, and MDPI. The databases were last searched on 29 October 2022. The inclusion criteria were as follows: (I) Original research, (II) Written and published in English, (III) Published between 1 January 2018 and 1 October 2022, (IV) Online access to and download of the work, made possible by the accessibility of the library of Lazarski University in Warsaw, (V) A complete DC/TMD or RDC/TMD study conducted, (VI) Subjects aged at least 18 years, (VII) Study conducted on human participants. Works excluded from this systematic review comprised studies meeting the following exclusion criteria: (I) Incomplete, absent, or modified version of DC/TMD or RDC/TMD protocol, (II) Systematic or narrative review, (III) Case reports, (IV) Study group size less than 100 participants, (V) Studies focusing exclusively on individual symptoms such as bruxism or headaches, which were not diagnosed as TMD, (VI) Absence of data analysis. Inclusion and exclusion criteria are presented in Table 1.

### 2.2. Search Strategy

An electronic search was conducted across Medline Complete, PubMed, and MDPI. The last database review took place on 29 October 2022. Boolean operators (OR and/or AND) combined with keywords were employed in each database search to achieve optimal results. The following keywords were used: ‘factors’, ‘causes’, ‘influences’, ‘TMD’, ‘temporomandibular disorder’, ‘temporomandibular disorders’, ‘DC/TMD’, and ‘RDC/TMD’. The following keywords were dropped in the last phase of the search: “factors”, “causes”, and “influences”, to obtain as many relevant papers as possible. In each of the searches, keywords were combined with the Boolean operator (OR and/or AND). A time restriction was imposed in all databases, limited to papers published between 2018 and October 2022. In the Medline Complete database, filters such as “find all search terms”, “use equivalent topics”, “full text”, “English language”, and “adults only” were utilized in the ‘advanced search’ option. In the ‘clinical questions’, ‘publication type’, and ‘gender’ options, no filters were applied. In the MDPI and PubMed databases, no additional limitations were applied. The search strategy is presented according to the PICO (patient/population, intervention, comparison, and outcomes) strategy in Table 2.

### 2.3. Selection and Data Collection Process

In phase one, two researchers (JM, JKG) independently performed a blind analysis of papers available in three databases based on titles and abstracts (TiAb screening). The papers that met the inclusion criteria were printed in full and analyzed manually in more detail by the first author (JW) (phase two). At this stage, the exclusion criteria were taken into account. Those papers that qualified in phase two proceeded to phase three, which consisted of a thorough analysis of the full texts by all reviewers (JM, JW, JGK) and a discussion of the resulting disagreements.

### 2.4. Data Items

The first author (JW) collected data from the selected papers, which was later verified for content validity and integrity by the third author (JM). The extracted information includes the following: first author’s name, year of publication, title, study group and its characteristics (gender, age, country of origin, any special features), research tools utilized, presence of control group, and outcomes associated with TMD etiology.

### 2.5. Study Risk of Bias Assessment

The risk of bias (RoB) was assessed independently by 2 authors (JW) and (JM) and discussed in a meeting with the whole team. The Joanna Briggs Institute (JBI) analytical cross-sectional studies’ Critical Appraisal Tool [20], consisting of 8 categories scored as ‘yes’, ‘no’, ‘unclear’, and ‘not applicable’, was used for the assessment. The categories assessed were clearly defined criteria for inclusion of the group in the study; study subjects and the setting described in detail; exposure measured in a valid and realistic way; objective, standard criteria were used for measurement of the condition; confounding factors were identified; outcomes were measured in a valid and reliable way; appropriate statistical analysis was used.

The “low risk of bias” group included papers with more than 85% “yes” responses, i.e., a maximum of 1 “no” response or 1 “unclear” response. The ‘moderate risk of bias’ group included papers with a maximum of 2 “no” answers or 2 unclear answers, i.e., papers with 62.5–75% “yes” answers. The final group, with a high risk of bias, included papers with less than 62.5% “yes” responses.

## 3. Results

### 3.1. Study Selection

Electronic searches were conducted using MDPI, Medline Complete, and PubMed databases. Two methods were used, depending on the selected keywords. Firstly, with the following keywords: ‘factors’, ‘causes’, ‘influences’, ‘TMD’, ‘temporomandibular disorder’, ‘temporomandibular disorders’, ‘DC/TMD’, ‘RDC/TMD’, 505 papers were acquired. Secondly, the search was narrowed by limiting the keywords to ‘TMD’, ‘temporomandibular disorder’, ‘temporomandibular disorders’, ‘DC/TMD’, and ‘RDC/TMD’, which received a total of 973 papers. A comprehensive review was conducted on 1478 papers. After eliminating duplicates, 1060 papers were assessed via abstract and title based on inclusion and exclusion standards. In-depth analysis was performed on 19 papers, with the last 8 being excluded. Eventually, this literature review included 11 papers [21,22,23,24,25,26,27,28,29,30,31]. The study selection process is shown in Figure 1.

### 3.2. Study Characteristics

The selected papers comprise studies conducted on adult patient populations from diverse countries, including Portugal, Finland, Poland, Germany, Italy, Slovenia, Brazil, and China. In 11 of the inclusions, the study group was a total of 8585 people (3357 subjects were assessed by RDC/TMD; 5228 were assessed DC/TMD), (3076 men, 5509 women). Three [21,28,31] of the papers selected were population-based studies, involving 5188 participants.

In addition to the DC/TMD or RDC/TMD questionnaires, two papers [21,22] used the Oral Behavior Checklist (OBC) to assess oral parafunctions and habits that can contribute to various oral health issues [32]. Three papers [21,22,23] used the Patient Health Questionnaire-9 (PHQ-9), a widely used questionnaire based on the Diagnostic and Statistical Manual of Mental Disorders (DSM-4) criteria, to screen and assess the severity of mental disorders [33]. The GAD-7 questionnaire is a tool used to measure the severity of generalized anxiety disorder [34]. It has been referenced in two papers [22,23]. The SCID (Structured Clinical Interview for DSM-III-R) is an instrument used to evaluate 33 commonly diagnosed Axis I DSM-III-R disorders in adults [35]. It has been referenced in two papers [24,25]. The Silness and Loe Plaque Index is a dental assessment tool that evaluates the presence and extent of dental plaque, providing insights into oral hygiene [36]. It was used in one paper [24]. The pericranial tenderness score (PTS), masticatory muscle tenderness score (MTS), and cervical muscle tenderness score (CTS) are components of an assessment tool for myofascial pain, specifically in the pericranial, masticatory, and cervical muscles [37,38]. Consecutively, PTS, MTS, and CTS were used in one paper [24]. The Oral Health Impact Profile (OHIP) is a questionnaire that assesses the impact of oral health on an individual’s quality of life [39]. It was used in one paper [22]. The Patient Health Questionnaire-15 (PHQ-15) is commonly employed as an accessible screening tool for somatization syndromes across various healthcare environments [40]. The Pittsburgh sleep quality index (PSQI) is a questionnaire used to assess sleep quality and disturbances over one month [41]. It was used in two papers [22,26]. These tools were each used in one paper [22]: The visual analog scale (VAS), graded chronic pain scale (GCPS), and Symptom Checklist-90 (SCL-90). These are tools to assess the subjective perception of acute and chronic pain [42], categorize and measure the severity of chronic pain [43], and assess a broad range of psychological problems and symptoms [44], respectively. The Fonseca Anamnestic Index (FAI) is a questionnaire used to evaluate the severity of temporomandibular disorders (TMD) based on the patient’s symptoms [45]. It was utilized in a single study [26]. The characteristics of the study are shown in Table 3.

### 3.3. Risk of Bias in Studies

Two papers [27,28] attained a ‘high risk of bias’ rating as they obtained a minimum of three responses other than yes, whereas one paper [22] received a “moderate risk of bias” rating. The criteria for which papers most often received a negative response were “strategies to deal with confounding factors” (which received five negative responses) and “clearly defined criteria for inclusion in the sample” (which received four negative responses and one unclear response). In the case of the strategy criterion, such a low score may have been due to the lack of clear guidelines on how to structure the work. This factor may also have been an indirect cause of the result for the inclusion criteria, as it should be noted that it was the exclusion criteria rather than the inclusion criteria that were most often described in detail.

It is noteworthy that high scores were attained in certain criteria, namely “appropriate statistical analysis was used”; “outcomes were measured in a valid and reliable way”; and “objective, standard criteria were used for measurement of the condition”, all of which achieved 100% positive responses. These scores may have been influenced by the inclusion and exclusion criteria of the study, which necessitated accurate, comprehensive, and suitable analysis for the study to be included in this literature review. Risk of bias (RoB) results are presented in Table 4.

### 3.4. Results of Individual Studies

The selected papers examined the following factors:Biological: sex, age, self-reported health condition, genetic mutations, oral parafunction, occlusion, condylar symmetry, skeletal divergence, extraction of teeth, orthodontic treatment, bruxism, fibromyalgia, migraine/headache, gastrointestinal disease, rheumatic disease, thyroid disease,Sociological: education, employment status, marital status, living conditions, socioeconomic status,Psychological: somatization, sleep quality, anxiety, depression, stress, chronic pain.

The data collected on the factors are presented in Table 5.

#### 3.4.1. Gender

The relationship between gender and the prevalence of TMD was examined in nine papers [21,22,23,24,25,26,28,29,30]. Seven papers [21,22,23,24,26,28,30] (total study group n = 3058) demonstrated a statistically significant association between the prevalence of TMD and female gender, while statistical significance was not established in two papers [25,29] (total study group n = 233).

#### 3.4.2. Age

Seven [21,22,24,25,26,29,30] studies analyzed a group of 3008 individuals to assess the correlation between age and TMD. Two studies [26,29] demonstrated a significant statistical association between TMD and increasing age (study group n = 1085), while in the remaining five studies [21,22,24,25,30], this factor was deemed statistically insignificant (study group n = 1923).

#### 3.4.3. Depression

Depression’s role as an etiological factor in TMD was analyzed in five papers [22,23,24,25,28], which collectively studied 2647 patients. All of these studies confirmed a statistically significant relationship between the two conditions.

To evaluate depression, the questionnaires utilized included the author’s questionnaire, GAD 7, PHQ-9, PHQ-15, and SCID I.

#### 3.4.4. Oral Parafunction

Oral parafunction as a risk factor of TMD, was studied in four papers [21,22,24,25], reaching a total of 1823 patients studied. The correlation proved statistically significant in two papers [21,22], a total of 1490 subjects. Two papers [24,25] showed no such correlation (n = 333 subjects). Given the data, the connection between oral parafunction and TMD was evidenced in 81% of the participants.

#### 3.4.5. Anxiety

An examination of anxiety as a contributing factor to TMD was conducted across three studies [22,23,24] involving a total of 578 patients. In all three studies, there was a statistically significant correlation found between the presence of TMD and anxiety.

The GAD-7 and SCID-1 were utilized to assess anxiety in patients.

#### 3.4.6. Somatization

Three papers [22,24,25] analyzed the correlation between TMD and somatization within a sample size of 442 participants. Out of these, two papers [22,24] verified a significant statistical correlation after studying 333 subjects, while one paper [25] did not report similar findings.

The SCID-1, PHQ-15, and SOM tools were employed to examine the occurrence of somatization among patients.

#### 3.4.7. Sleep Quality

Sleep quality was assessed in three papers [22,26,27] featuring 2713 subjects. All three papers confirmed a statistically significant relationship.

The PSQ1 and SAC were utilized to investigate sleep quality.

#### 3.4.8. Anatomy and Factors Affecting Anatomy (Orthodontic Treatment)

Three papers [24,30,31] on this topic were identified, with a total of 2169 subjects.

One study [31] analyzed sagittal molar and canine relationships, asymmetries, and midline shifts in TMD patients. The study found statistically significant relationships between TMD and several occlusal abnormalities, which included cusp-to-cusp class II molar relationships, midline shift, canine asymmetry, and missing first molars or canines (study group n = 1845).

The second study [24] aimed to examine the impact of orthodontic treatment on TMD development (n = 224). Although the study found no significant relationship between orthodontic treatment and TMD, pain scores for myofascial pain syndrome appeared to worsen during the treatment. Nevertheless, after completing the treatment, pain showed an improvement.

The third study [30] investigated the correlation between TMD and condylar asymmetry and the growth’s divergence pattern (n = 100). This study demonstrated a strong correlation between condylar asymmetry and hyperdivergence facial growth patterns.

#### 3.4.9. Social Factors

This literature review examines the social factors of education, employment status, marital status, living conditions, and socioeconomic status. The review analyses two papers, one on employment [28] and the second [22] on other factors.

The results indicate a significant statistical relationship only for employment status, which was examined in one paper [28] with a sample size of 1962.

There was no significant correlation found between education (two papers [22,28], n = 2071), living conditions (one paper [28], n = 1962), socioeconomic status (one paper [28], n = 1962), and marital status (two papers [22,28], n = 2071).

#### 3.4.10. Other Factors

Other factors were explored individually in single papers, and they revealed a significant statistical association with self-reported health conditions [28] (n = 1962), fibromyalgia [28] (n = 1962), gastrointestinal disease [28] (n = 1962), migraine/headache [28] (n = 1962), osteoarthritis [28] (n = 1962), rheumatic disease [28] (n = 1962), thyroid disease [28] (n = 1962), and genetic mutation [29] (n = 142).

There was no statistical correlation observed among variables including chronic pain [22] (n = 109), and stress [25] (n = 109).

### 3.5. Results of Synthesis

Due to the diverse nature of the papers and the various methods implemented to study the factors, a high degree of heterogeneity is exhibited among the papers, rendering comparisons challenging. Of the eleven papers chosen for appraisal, three are population-based studies, with successive observations conducted on cohorts of n = 1381 [21], n = 1845 [31], and n = 1962 [28]. The systematic review identified a significant correlation between female gender and temporomandibular disorders (TMD) in seven out of nine studies. TMD was linked to increasing age in two out of seven studies. Psychological factors, such as depression and anxiety, exhibited consistent statistically significant associations with TMD, as documented in five and three research papers, respectively. Oral parafunction displayed varied correlations, with specific parafunctions like clenching or grinding teeth exhibiting significance in two out of four studies. Sleep quality showed a consistent association with TMD in three studies. Occlusal abnormalities, orthodontic treatment, and condylar asymmetry were linked to TMD in the context of papers that explored anatomical features. Other individual factors, such as employment status, health conditions, and genetic mutations, demonstrated noteworthy associations with TMD in individual studies. Nevertheless, statistical correlations were not observed for variables such as marital status, education, chronic pain, bruxism, and stress, which were also examined as factors in the individual studies. Table 5 includes data on the analyzed factors.

## 4. Discussion

The factors influencing TMD can be classified into three main groups: organic, psychological, and social factors. Organic factors can be further divided into central and peripheral factors [46]. Peripheral factors are caused by disorders associated with abnormalities of the peripheral nervous system and other tissues. These include ongoing inflammatory processes, autoimmune diseases, organ abnormalities, and past trauma. Central factors include abnormalities related to the central nervous system, such as psychological impairment, neuropathic pain, or selected sleep disorders. Other organic factors include gender, age, and genetic disorders. Social factors consist of occupation, economic status, and social conditions, among others [47]. In 1977, George Engel proposed the biopsychosocial model as a multidimensional approach to disorders [48]. This model considers various factors that can influence the development of TMD, which is a major diagnostic challenge to investigate thoroughly. The DC/TMD protocol is currently the most versatile and comprehensive tool for a multi-specialist approach to diagnosing TMD incorporating the biopsychosocial model. Its use is a crucial inclusion criterion for this work.

The DC/TMD protocol is currently the most widely used by clinicians worldwide. It enables standardization of diagnosis and provides a basis for objective data comparison. The DC/TMD provides a practical classification of TMD, which distinguishes various disorders, including myalgia, local myalgia, myofascial pain with spreading, myofascial pain with referral, arthralgia, headache attributed to TMD, disc displacement with reduction, disc displacement with reduction and limited opening, disc displacement with reduction and without limited opening, degenerative joint disease, and subluxation [49]. This analysis also utilizes RDC/TMD (Research Diagnostic Criteria for Temporomandibular Disorders), the prototype of DC/TMD, as a keyword. RDC/TMD was introduced in 1992 to standardize the classification of a group of patients affected by TMD. It underwent refinement until 2014, when the current version, DC/TMD, was introduced. A study by Samuel F. Dworkin et al. examined the reliability, validity, and clinical utility of the Research Diagnostic Criteria for Temporomandibular Disorders Axis II Scales [50]. The study found that the psychometric properties of the RDC/TMD are well-suited for the thorough evaluation and treatment of individuals presenting with Temporomandibular Disorders (TMD) [50].

This literature review aims to identify factors that may impact the development of TMD.

### 4.1. Gender

Research suggests that TMD may occur up to two to three times more frequently in women than in men [51]. There are several theories to explain this gender imbalance in TMD prevalence. One theory is that women tend to be more attentive to distressing symptoms and seek medical help more frequently than men [52], which may result in more frequent TMD diagnoses in this group. Another theory is the estrogen theory. Endogenous estrogens and their cyclical fluctuations can influence several factors that may eventually lead to TMD, such as gingivitis, periodontal disease, condylar fibrocartilage, protease activity, and estrogen signaling [53]. In a 1997 study, LaResche et al. [54] found an association between taking exogenous estrogens in hormone replacement therapy and an increased incidence of TMD symptoms, compared to women who did not receive such therapy. Seven [21,22,23,24,26,28,30] out of nine papers that were examined found a statistically significant correlation in this analysis. The remaining two papers [25,29] also showed a correlation, but it was not statistically significant. The authors themselves reported that the lack of statistical significance may be due to the selection of specific patient groups.

### 4.2. Age

Among the etiological factors of TMD, age is often cited as a potential risk factor. However, a comparison of the papers studied does not provide sufficient evidence to draw a similar conclusion. Out of the seven papers reviewed, only two [26,29] confirmed this relationship. Proportionally, the total study group in which such a relationship was confirmed, compared to the group in which no statistical significance was demonstrated, is 1:1.77 (n = 1085, n = 1923). Such a result is in contrast to the study of Gary D. Slade et al. [47], which showed that the site-adjusted incidence rate of first-onset Temporomandibular Disorder (TMD) increased with age across the entire cohort of 2737 individuals. The incidence rate rose from 2.5% per annum in the 18 to 24-year-old group to 4.5% per annum in the 35 to 44-year-old group [47].

### 4.3. Depression

Depression and anxiety were the only factors that were statistically significant in all of the papers studied. V. Aggarwal et al. [55] demonstrated that anxiety contributes to the development of chronic orofacial pain, while L. Simoen et al. [23] recommended that questionnaires assessing anxiety in patients be included in the diagnosis of orofacial pain associated with TMD. It is important to note that in numerous research papers, anxiety is often associated with depression, and the findings are frequently reported together. However, this review presents separate results for depression and anxiety, enabling accurate conclusions to be drawn for each factor. According to S. Kindler et al. [56], “depressive symptoms were more strongly associated with joint pain compared to muscle pain, while anxiety symptoms were more strongly associated with muscle pain compared to joint pain”. Research suggests that anxiety can cause muscle hyperactivity, resulting in muscle fatigue and compensatory behavior [57]. This can lead to degenerative arthritis, chewing disorders, and disharmonious occlusion, all of which contribute to the development of TMD. The relationship between TMD and depression and anxiety has been debated for years, but is now considered to be two independent disorders: C. Stavrakaki et al. [58] state that, “anxiety and depression are classified as separate disorders—clinically through DMS-III and statistically through discriminant function analysis”. Several main streams explain the relationship between depression and TMD. The three main explanations are that depression is the result of a pain disorder; TMD is the result of depression (also known as ‘masked depression’); and depression and pain result from a more central disorder [59]. Gallagher et al. [60] conducted a study on 106 women and found that 41% of patients diagnosed with TMD had a history of major depression. A study conducted on a cohort of students found that both depression and perceived stress and mood are risk factors in the development of TMD [61].

### 4.4. Oral Parafunction

Oral parafunctions are abnormal activities that involve the oral structures, such as the jaw, teeth, and surrounding tissues, but are not part of typical functional movements, such as talking or chewing. These behaviors can adversely affect oral health and may lead to various dental problems. Typical oral parafunctions include onychophagia (nail biting), cheek biting, lip biting, tongue thrusting, tongue chewing, and mouth breathing. Previously, the parafunction groups also included bruxism, which is now considered a behavioral activity. With a prevalence of up to 90% in the general population, bruxism and clenching are the most common oral activities [62]. Two studies [21,22] have demonstrated a statistically significant association between oral parafunctions and TMD. Both studies employed the Oral Behavior Checklist (OBC), and one of them also used the Oral Health Impact Profile Questionnaire (OHIP) [22]. The OBC is a questionnaire that patients complete to evaluate oral and orofacial parafunctional disorders. The OHIP questionnaire is utilized to assess the effect of oral conditions on an individual’s quality of life, encompassing physical, psychological, and social well-being [63]. In contrast, the two papers that failed to confirm the correlation between oral habits and quality of life employed questions from the RDC/TMD questionnaire and their questionnaires, which relied on YES/NO responses. It is important to note the relationship between the use of tools specifically designed to assess parafunctions and their association with TMD. The lack of a statistically significant correlation between the two may be related to the use of non-specific tools to assess oral habits.

### 4.5. Somatization

Somatization was found to have a significant statistical relationship in two [22,24] out of the three papers in which it was studied. This mental disorder involves expressing emotional distress through physical symptoms. Increased pain sensation is a common symptom of somatization, as supported by the literature. Wilson et al. [64] demonstrated a correlation between elevated levels of somatization and TMD pain. Rehm et al. [65] also reached similar conclusions. The study found that individuals with Temporomandibular Disorder (TMD) who experience muscle pain and arthralgia/osteoarthritis have higher pain intensity compared to those with disc displacement and lower pain intensity. This higher pain intensity is directly associated with secondary depression and nonspecific symptoms/somatization induced by TMD. According to Canales et al. [66], individuals with Temporomandibular Disorder (TMD) typically display an emotional profile that is characterized by low levels of disability, significant impairment related to pain intensity, and moderate to elevated levels of somatization and depression. Dworkin et al. [67] emphasized the importance of distinguishing between somatization as a mental disorder and somatization as a character trait.

### 4.6. Sleep Quality

Sleep quality was found to be consistently associated with pain in all reviewed papers [22,26,27]. P.H. Finan et al. [68] hypothesized that sleep is a reliable predictor of pain and can contribute to the development of chronic pain in joints, which may also be relevant to the development of TMD. The impact of sleep quality on pain may be associated with the pathways of serotonergic and dopaminergic neurotransmission. Furthermore, sleep deprivation can cause elevated cortisol and stress levels, leading to significant daytime sympathetic nervous system stimulation and increased muscle activity. A separate study conducted on a group of adolescents revealed a high incidence of sleep deprivation among college students, which increased the risk of TMD symptoms [69]. Although sleep showed a statistical relationship in each of the papers in which it was studied, it should be noted that only three such papers were included in this review. Despite the rather large group in which this factor was examined (n = 2,713), the potential risk of bias must be taken into account.

### 4.7. Anatomy and Factors Affecting Anatomy (Orthodontic Treatment)

This group included occlusal relations, skeletal relations, and changes in occlusal contacts associated with orthodontic treatment. The relationship between TMD and occlusion is still a matter of debate. Occlusion, in the context of dentistry, denotes the dynamic relationship between the maxillary and mandibular teeth during functional and parafunctional activities, encompassing biting, chewing, and other masticatory functions. Robert J.A.M. de Kanter et al. [70] discuss occlusion in four aspects. Firstly, it refers to the anatomic or ‘orthodontic’ jaw relation, including the Angle classification. Secondly, it refers to the static contact between the upper and lower teeth. Thirdly, it involves dynamic contact during functional activities, such as cuspid guidance versus group function, articulation, and identification of occlusal interferences. Finally, occlusion also applies to prosthetic classifications, which distinguish between complete and incomplete dentition, as well as fixed or removable prosthetics.

Three papers were found that addressed this issue [22,30,31], each examining a different aspect of the anatomical features. The most relevant paper seems to be that of the Finnish authors [31], due to the large study population, confirming the relationship between the occurrence of TMD and such occlusal disorders as sagittal molar class II cusp-to-cusp relationships, asymmetries, midline shift, and missing teeth. Also, work investigating skeletal divergence and condylar asymmetry [30], confirms the statistical relationship between hyperdivergent skeletal pattern and condylar asymmetry as a risk of TMD. The results of this study provide a field for discussion. Manfredini et al. state that there is insufficient basis to hypothesize a significant role for dental occlusion in the pathophysiology of Temporomandibular Disorders (TMDs) [71].

In the past, it was commonly believed that occlusal disorders affecting the position of the condylar process in the TMJ were the main cause of TMD complaints [72]. This led to TMD therapy being primarily based on occlusal adjustment. However, there is no evidence to support the effectiveness of this therapy compared to a placebo in treating TMD [73]. On the other hand, there is evidence to suggest a correlation between vertical growth patterns and TMD, as confirmed by several studies [71,74]. The third study [24] investigated the impact of orthodontic treatment on TMD. The authors observed that during active tooth shifting, TMD worsened in 13.3% of patients, primarily in the form of myofascial pain. However, this was significantly associated with factors such as gender, anxiety, or depression. Furthermore, there was a higher frequency of these complaints among Angle’s class II patients, although the correlation was not statistically significant. The study results confirm the dominant role of psychosocial factors over occlusal factors in TMD. However, there is a lack of research on the relationship between malocclusion and TMD based on DC/TMD diagnosis. This paper only pertains to a group of orthodontic patients, and there is no comparison to the development of TMD symptoms in a control group with similar parameters related to gender, age, and orthodontic needs, and without orthodontic therapy.

### 4.8. Stress

Stress is believed to be a contributing factor in the development of TMD. This may be due to increased cortisol levels [75], which can lead to heightened muscle activity. Such activity can negatively impact proper muscle function, resulting in altered joint mechanics. Furthermore, increased muscle activity may contribute to the development of arthritis, which can cause joint damage [56]. Abnormalities in the muscles and joints in the temporal region can disrupt homeostasis in the trigeminal nerve. These abnormalities affect the balance of neurotransmitter secretion, such as serotonin and catecholamines [76]. In this literature review, there was only one paper referring to a study of the correlation between stress and TMD, and it showed the absence of such a correlation [25]. According to other sources, however, such a correlation does occur. K. Staniszewski et al. [77], found significantly higher levels of stress in the TMD group than in the control group, and Gui Maísa Soares et al. [78] advocate that chronic TMD is associated with psychological distress and pain amplification characteristics. In recent years, one factor that may have influenced stress levels in the population, and thus TMD, has been the coronavirus pandemic. Also, such a thesis finds support in the literature: Sabina Saccomanno et al. [79] conducted an online survey of patients during the last phase of the lockdown in Italy (April–May 2020); in patients who reported experiencing orofacial pain, 51.4% indicated a deterioration of symptoms in the preceding month. Notably, a substantial 94.7% of these individuals (35 subjects) attributed the escalation of pain to the stressful conditions during the lockdown and its associated consequences. However, the relationship between TMD and stress as a factor is still debatable. R. Ohrbach et al. [80], proposed a hypothesis in which stress and awake oral parafunction admittedly influence the occurrence of myofascial pain, but must still coexist with other factors.

### 4.9. Social Factors

This literature review examines the biopsychosocial factors contributing to TMD. The social factors analyzed include education, marital status, employment status, living conditions, and socioeconomic status. The statistical analysis revealed that individuals who were unemployed or retired exhibited a higher prevalence of pain-related symptoms compared to those who were currently employed [28]. The study found no correlation between other factors. However, the authors themselves suggest that this may be due to the homogeneity of the study group, where the standard of living is at a similar level. Gary D. Slade [47] conducted a study on the indicators and manifestations of initial onset temporomandibular disorder and the sociodemographic factors that influence its onset. The study found that the incidence of Temporomandibular Disorder (TMD) was associated with a subjective measure of material satisfaction, but not with objective indicators of socioeconomic status such as education and income. It is worth emphasizing that the assessment of social factors is difficult, due to the differences in the study groups resulting from the standard of living in each country and the level of public awareness that this entails, as well as the lack of uniform tools that would assess the factors in question as objectively as possible.

## 5. Limitations

The present study has limitations. One of these limitations is the lack of a meta-analysis performed on the results obtained. This is due to the heterogeneity and the design of the papers that were studied. The selected papers investigated the correlation or lack of correlation between TMD and a given etiologic factor. Despite consulting with master statisticians, it was not possible to perform a meta-analysis. In the future, it may be worthwhile to consider the feasibility of conducting studies that would enable a proper meta-analysis. Additionally, the number of papers that examined individual factors was relatively small, with some factors being studied by only one paper. This limitation is due to the inclusion and exclusion criteria of this review, which required a minimum of n = 100 study subjects. To improve the search for factors that may influence TMD, it is recommended to expand the scope of the search by including more databases.

One of the main limitations of the study, partially self-imposed, is the inclusion criteria of at least 100 participants. The authors aimed to select papers examining large groups of people to reduce the risk of statistical error, as the paper is not based on a statistical study. However, this approach resulted in the rejection of many papers that could potentially describe real TMD factors. To further investigate this topic, it may be beneficial to consider reducing the size of the study groups and conducting statistical analyses on the literature review that was performed.

One of the potential limitations is that the search was conducted in a relatively small number of databases, and using primarily university library resources. In the future, it may be beneficial to conduct searches in a wider range of databases. To ensure comprehensive research, it may be beneficial to explore a wider range of databases in the future.

Another limitation due to the inclusion and exclusion criteria of this work is language. Only papers published in English were included in this review, which could potentially exclude relevant studies conducted in other languages. Future studies would also need to review papers published in other languages. The choice of the DC/TMD and RDC/TMD surveys should also be considered as a potential risk of bias of this work, due to the focus of these questionnaires on psycho-sociological disorders. The review’s findings are primarily limited to Europe, Brazil, and China, which may impact the outcome due to ethnic differences between populations. It is possible that this limitation was caused by the inclusion of papers in English only.

## 6. Conclusions

The study group size and the number of papers confirming the statistical significance of a given factor suggest that the most significant factors influencing the development of TMD are female gender, poor sleep quality, depression, occlusion, oral parafunction, anxiety, somatization, self-reported health condition, fibromyalgia, migraine/headache, gastrointestinal disease, thyroid disease, osteoarthritis, rheumatoid disease, and employment status.

Given the findings that demonstrate the influence of psychological factors on TMD, it is essential to comprehend how psychological dysfunctions can be identified more effectively in patients at risk of TMD development. Efficient early intervention programs necessitate development, and a comprehensive management approach involving the collaboration of various professionals should be implemented.

The DC/TMD and RDC/TMD questionnaires, as diagnostic tools, not only allow the diagnosis of the disease but also allow the analysis of the factors involved. Therefore, this questionnaire can be a useful tool not only for researchers but also for clinicians in their daily practice.

## Figures and Tables

**Figure 1 healthcare-12-00575-f001:**
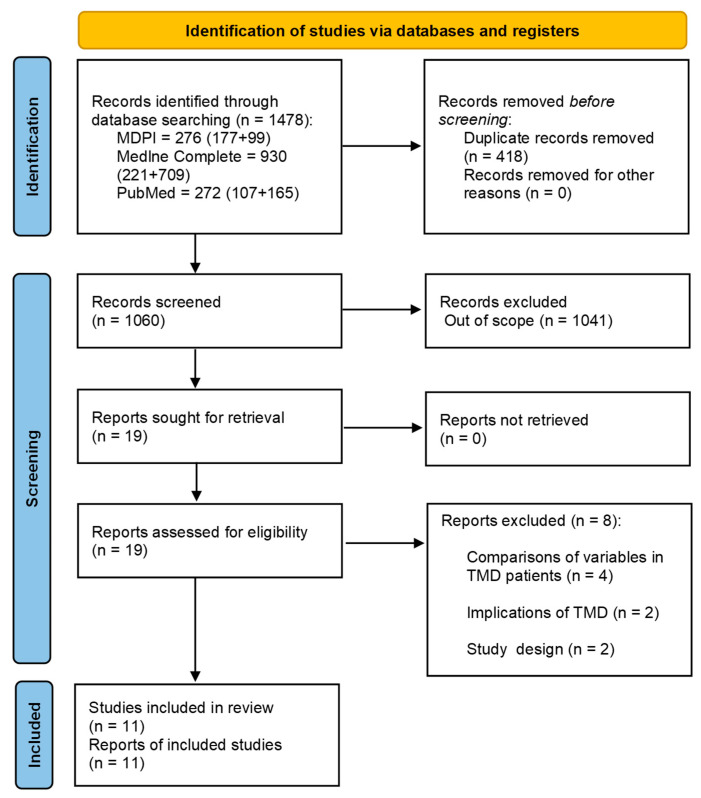
The flow diagram of the included literature searching strategy.

**Table 1 healthcare-12-00575-t001:** Inclusion and exclusion criteria.

Inclusion Criteria	Exclusion Criteria
Original research	Incomplete/absent/modified DC/TMD or RDC/TMD
Articles in English, published between 1 January 2018 and 1 October 2022	Systematic/narrative review
Full DC/TMD or RDC/TMD	Case reports
Participants at least 18 years old	Study group < 100 participants
Only human participants	Symptoms of TMD, defined as TMD
Full online access granted by the library of Lazarski University in Warsaw	Absence of data analysis

**Table 2 healthcare-12-00575-t002:** PICO strategy used in the search strategy.

PICO Elements	Keywords	Search Items	Search Strategies
P: Patients or Population	Patients diagnosed with TMD using DC/TMD or RDC/TMD	Patients suffering from TMD	Temporomandibular disorder OR temporomandibular disorders OR TMD AND DC/TMD OR RDC/TMD
I: Intervention	Factors contributing to the etiology of TMD, as measured by standardized questionnaires or other diagnostic methods	Factors impacting TMD	Factors OR influences OR causes
C: Comparison	TMD patients and general population; TMD patients and healthy control groups; comparison within TMD patients		
O: Outcome	Factors impacting TMD		

**Table 3 healthcare-12-00575-t003:** PICO.

Study First Author, Year	Title	Population (P)	Intervention (I)	Statistical Analysis (SA)	Comparison (C)	Outcomes (O)
Cláudia Barbosa, 2021 [21]	Are oral overuse behaviors associated with painful temporomandibular disorders? A cross-sectional study in Portuguese university students	N = 1381 (M: 339, F: 1042), range: 18–67, mean age: 21, population: Portuguese University Students	RDC/TMD;	Univariate associations between categorical variables were tested using chi-square tests.	nonapplicable	Painful TMDs were significantly associated with both low- and high-frequency oral parafunctional behaviors. (*p* < 0.001)
				Univariate logistic regression was used to assess the relationship between individual OBC items and OBC sum score categories with painful TMDs and other TMDs.		Other TMDs were associated mainly with the high-frequency behavior of clenching or grinding teeth during sleep. (*p* = 0.03)
			Oral Behavior Checklist (OBC);	*t*-tests were used to compare the OBC sum score between dichotomous groups (sex and age).		The model retained specific behaviors, such as holding or jutting the jaw forward/to the side, clenching or grinding teeth during sleep, grinding teeth when awake, holding the jaw in a rigid position, leaning the jaw, and sustained talk, as independent variables associated with painful TMDs.
				ANOVA followed by Scheffe post hoc test was used among diagnostic groups (TMD-free, other TMDs, and painful TMDs).		Painful TMDs were more prevalent in females (30.7%) than males (19.5%). (Mann–Whitney test, *p* < 0.001)
			Patient Health Questionnaire (PHQ-9);	Multivariable binary logistic regression models were used to test the association between individual oral parafunctional behaviors, OBC sum score categories, and painful TMD subtypes.		Females, younger adults, and those with painful TMDs had higher oral behaviors checklist (OBC) scores. (*p* = 0.001, *p* < 0.001, *p* < 0.001, respectively)
						High OBC scores were associated with myalgia (*p* = 0.002), arthralgia (*p* = 0.001), and combined myalgia and arthralgia (*p* ≤ 0.001).
						Low OBC scores were only associated with combined myalgia and arthralgia (*p* = 0.001).
Päivi Jussila, 2018 [28]	Association of risk factors with temporomandibular disorders in the Northern Finland Birth Cohort 1966	N = 1962, (M: 912, F: 1050), range: born in 1966 (46 y.o), population: Finland	DC/TMD;	Chi-square Test	non applicable	Female gender and self-reported poor/fair health conditions were strongly associated with pain-related symptoms and clinical signs of Temporomandibular Disorders (TMD) (*p* < 0.000).
						Subjects with poor or fair health conditions had more pain-related TMD symptoms and pain in the masticatory muscles and TMJs (*p* < 0.000).
			Additional questionnaires about comorbid factors: gender, employment, self-reported health conditions, depression, fibromyalgia, gastrointestinal disease, migraine headache, osteoarthritis, rheumatic disease, thyroid disease			Subjects not working or retired had a numerically higher prevalence of pain-related symptoms in the temples, TMJs, face, or jaw compared to those currently working (*p* > 0.05).
						Diagnosed depression, migraine, fibromyalgia (FM), rheumatic disease, and general osteoarthritis showed statistically significant associations with pain-related TMD symptoms (*p* < 0.05).
						Migraine, FM, rheumatic disease, and general osteoarthritis were also associated with pain-related TMD symptoms during maximal mouth opening or chewing (*p* < 0.05).
				Fisher’s exact test used along with the chi-square test		Thyroid disease and gastrointestinal disease were associated with pain in the masticatory muscles and pain in the TMJs (*p* < 0.05).
						Self-reported sleep apnea (diagnosed by a physician) was associated with clicking in the TMJs (*p* = 0.029).
						Current smoking or use of snuff had an association with clinical TMD signs (*p* = 0.021).
				Statistical significance was determined at *p* < 0.05.		General health problems (depression, migraine, FM, gastrointestinal diseases, rheumatic disease, and general osteoarthritis) were strongly associated with TMD pain symptoms (*p* < 0.05).
						Diagnosed depression showed a strong association with pain on palpation in the masticatory muscles and pain in the TMJs (*p* < 0.05).
						Perceived stress and personal well-being were associated with diagnosed depression and pain on palpation in the masticatory muscles and TMJs (*p* < 0.05).
						Subjects not working or retired had more pain-related symptoms than those currently working (*p* > 0.05).
Bartosz Dalewski, 2021 [29]	COL5A1 RS12722 Is Associated with Temporomandibular Joint Anterior Disc Displacement without Reduction in Polish Caucasians	N = 124, (M: 20, F: 104), mean age: 32.36, population: Poland	DC/TMD;	Odds ratios (OR) were calculated in relation to the most frequent combination with 95% confidence intervals.	Patients with no TMD problems: N = 126 (M: 30, F: 96), mean age: 43.86, population: Poland	The COL5A1 marker rs12722 showed significant *p*-values, indicating differences in the frequencies of temporomandibular joint disc dislocation. (*p* = 0.0119)
			CBCT/MRI;	Pearson’s chi-square test was used to assess the significance of genotype distribution differences.		
				Logistic regression modelling was employed to analyze the influence of investigated SNPs on ADDwoR.		
			SWAB Genomic Extraction GPB Mini Kit	The Student’s *t*-test was used to determine age differences between groups.		Patients with the rs12722 genotype CT had an almost 2.4 times higher likelihood of disc dislocation (OR = 2.41) compared to those with the reference genotype TT (OR = 1). (0 = 0.0032)
				A chi-square test was used for sex distribution analysis.		Analysis of COL5A1 markers revealed a significant association for rs12722. Patients with rs12722 genotype CT had a 2.4 times higher likelihood of disc dislocation compared to TT (OR = 2.413, *p* = 0.003), while rs13946 genotypes showed no significant association.
				Statistical significance was set at *p* < 0.05.		Logistic regression confirmed the significant impact of the rs12722 CT allele on disc dislocation (OR = 2.413, *p* = 0.003). The rs13946 genotypes did not exhibit a significant effect in the logistic regression model.
Louis Simoen, 2020 [23]	Depression and anxiety levels in patients with temporomandibular disorders: comparison with the general population	N = 243 (M: 52, F: 191), mean age: 41; population: Germany	DC/TMD;	Kolmogorov–Smirnov Test (KS Test)	N = 5018 for PHQ-9; N2 = 5026 for GAD-7	Patients with pain attributed to TMD exhibited significantly higher scores on both PHQ-9 and GAD-7 compared to a general population sample. (*p* < 0.05)
				Chi-square Test		
			Generalized Anxiety Disorder Assessment (GAD-7)	Spearman’s rank correlation tests		
			Patient Health Questionnaire (PHQ-9)	Pearson correlation		19% of the study sample had a PHQ-9 score ≥ 10 (moderate depression), while 29% had a GAD-7 score ≥ 10 (moderate anxiety). In contrast, the reference population had lower percentages (7% and 6%, respectively).
				Results were considered statistically significant at *p* ≤ 0.05.		
Alessandro Ugolini, 2020 [24]	Determining Risk Factors for the Development of Temporomandibular Disorders during Orthodontic Treatment	N = 224 (M: 105, F: 119), mean age: 28; population: Italy	RDC/TMD;	Multivariate logistic regression analysis	No control group	Gender played a crucial role; women had 90% higher odds than men of developing TMD. (*p* < 0.01)
			Structured Clinical Interview For DSM-IV (SCID-I)	Odds ratios (ORs) were adjusted for age, sex, and the presence of anxiety or mood disorders.		There is a statistically significant relationship between the presence of TMD and anxiety, depression, and somatization (*p* < 0.01)
			Silness and Loe plaque index			
			Angle’s classification			
			Pericranial tenderness score (PTS)			
			Masticatory muscle tenderness score (MTS)			
			Cervical muscle tenderness score (CTS)			
Tadej Ostrc, 2022 [22]	Headache Because of Problems with Teeth, Mouth, Jaws, or Dentures in Chronic Temporomandibular Disorder Patients: A Case–Control Study	N = 109 (M: 17, F: 92), mean age: 35.07; population: Slovenia	DC/TMD;	Cohen’s d, a measure of effect size, was calculated to demonstrate the standardized differences between the two groups.	Patients with TMD, without headache because of problems with teeth, mouth, jaws, or dentures (HATMJD); M = 68 (M: 23, F: 45); mean age: 38.9	Simple logistic regression:
			Oral Health Impact Profile questionnaire (OHIP)	Simple logistic regression analysis		Female gender was significantly associated with TMD patients reporting HATMJD (*p* = 0.005)
			Patient Health Questionnaire (PHQ-9)	Multiple logistic regression analysis		Depression (*p* = 0.011), anxiety (*p* = 0.020), and physical symptoms (*p* = 0.001) were significantly associated with TMD patients reporting HATMJD.
			Generalized Anxiety Disorder (GAD-7 summary score)	Omnibus Test of Model		Oral behaviors (OBC summary score) were significantly higher in the group with HATMJD (*p* < 0.001).
			Patient Health Questionnaire-15 (PHQ-15)	Hosmer–Lemeshow test		Sleep quality (PSQI summary score) was significantly worse in the HATMJD group (*p* < 0.001).
			Oral Behavior Checklist (OBC)	Nagelkerke’s R-square		Multiple logistic regression:
			Pittsburgh sleep quality index (PSQI)	Odds Ratio Calculation		Female gender (*p* = 0.023), oral behaviors (*p* = 0.019), sleep quality (*p* = 0.021), and depression (*p* = 0.549) were identified as significant predictors of HATMJD.
				Statistical significance was set at *p* < 0.05		
Magdalena Osiewicz, 2019 [25]	Pain Predictors in a Population of Temporomandibular Disorders Patients	N = 109 (M: 22, F: 87), mean age: 33.2, range: 18–72; population: Poland	RDC/TMD;	Single-variable logistic regression analyses were performed to assess the association between various predictors and the TMD group (pain group vs. non-pain group)	No control group	Only gender (*p* < 0.063) and depression (*p* < 0.019) showed a significant correlation with TMD.
			Visual Analog Scale (VAS);			
			Graded Chronic Pain Scale (GCPS);	Variables removed until all retained variables showed *p* < 0.05.		
			Symptoms Checklist-90 (SCL-90);	A multiple logistic regression model was attempted, but only depression (DEP) remained a significant variable.		
			Cervical muscle tenderness score (CTS);	Odds ratios (OR) assessed for each variable.		
			Somatization Scale (SOM);	Nagelkerke’s R-square		
			Diagnostic and Statistical Manual of Mental Disorder (SCID I);			
Elisa Tervahauta, 2022 [31]	Prevalence of sagittal molar and canine relationships, asymmetries and midline shift in relation to temporomandibular disorders (TMD) in a Finnish adult population	N = 1845 (M: 857, F: 988), mean age: 46; range: 46; population: Finland	DC/TMD;	Pearson’s chi-square test (χ^2^-test)	Non applicable	Canine relationship and canine asymmetry were associated with pain in temples, TMJs, face, or jaw (*p* = 0.015, *p* = 0.039).
						In the multivariable model, the association between canine asymmetry and pain remained significant (*p* = 0.041).
						Limited mouth opening was more frequent in subjects with asymmetry in canine relationships (*p* = 0.040).
				Fisher’s exact test		A statistically significant difference in crepitus in TMJs was observed between groups of different molar relationships, with the half-cusp Class II group being most affected (*p* = 0.020).
			iTero 3D scanner;			In the multivariable model, half-cusp Class II remained significantly associated with crepitus in TMJs (*p* = 0.024).
						Half-cusp Class II was the most frequent bilateral molar relationship in females with disc displacement with reduction (15.8%, *p* = 0.043) and degenerative joint disease (26.3%, *p* < 0.001).
				Logistic regression analyses		The bilateral molar relationship was statistically significantly associated with disc displacement with reduction and degenerative joint disease (*p* = 0.034, *p* < 0.001, respectively).
						Females with one or more missing canines had significantly more myalgia and arthralgia compared to females with no missing canines (*p* = 0.014, *p* = 0.022).
			Self-reported general health questionnaire	Significance level set at *p* < 0.05.		Self-reported general health was significantly associated with pain symptoms, limited mouth opening, myalgia, and arthralgia (*p* < 0.001, *p* = 0.047, *p* < 0.001, *p* < 0.001, respectively).
						Mental health was associated with arthralgia (*p* = 0.049).
						Rheumatoid arthritis was associated with crepitus in TMJs and arthralgia (*p* = 0.002, *p* = 0.018).
						Gender was associated with TMD symptoms and signs (*p* < 0.001, *p* = 0.004, *p* = 0.024).
Maria Francesca Sfondrini, 2021 [30]	Skeletal Divergence and Condylar Asymmetry in Patients with Temporomandibular Disorders (TMD): A Retrospective Study	N = 100 (M: 34, F: 66); range: 18–30; population: Italy	DC/TMD;	Descriptive statistics, including mean, standard deviation, minimum, median, and maximum values, were calculated for all numerical groups.	Patients without TMD; N = 100 (M: 46, F: 54)	Patients with a TMD diagnosis showed significantly greater skeletal divergence with a higher SpPGoGn angle (*p* = 0.00155).
				A linear regression model for TMD was performed, adding age, sex, symmetry, and divergence as covariates.		A strong statistically significant difference in the condylar symmetry parameter was observed, with the TMD group having a much higher percentage of asymmetric condyles (*p* < 0.0001).
			X-ray Analysis;	Significance level set at *p* < 0.05 for all tests.		Regarding gender, a statistically significant difference was found between the two groups (*p* = 0.0444), while no difference in age was detected (*p* = 0.297).
Daniela D. S. Rehm, 2019 [27]	Sleep Disorders in Patients with Temporomandibular Disorders (TMD) in an Adult Population- Based Cross-Sectional Survey in Southern Brazil	N = 1643 (M: 561, F: 1082), range: 18–65; population: Brazil	RDC/TMD;	Student *t* test	No control group	Global sleep score, insomnia, nonrestorative sleep, schedule disorders, daytime sleepiness, sleep apnea, restlessness were the factors in which mean scores were significantly higher in TMD subjects compared to controls (*p* < 0.001).
						72.1% of TMD subjects had a global sleep disorder compared to 48.2% of controls (*p* < 0.001).
						37.5% of TMD subjects had insomnia compared to 14.7% of controls (*p* < 0.001).
				Pearson chi-square test		47.2% of TMD subjects had nonrestorative sleep compared to 18.2% of controls (*p* < 0.001).
			Sleep Assessment Questionnaire (SAQ)			60.0% of TMD subjects had sleep schedule disorders compared to 52.8% of controls (*p* < 0.01).
						26.6% of TMD subjects had daytime sleepiness compared to 16.2% of controls (*p* < 0.001).
						24.7% of TMD subjects had sleep apnea compared to 17.4% of controls (*p* < 0.001).
						TMD subjects showed a higher prevalence of restlessness (*p* < 0.001).
Adrian Ujin Yap, 2021 [26]	Temporomandibular disorder severity and diagnostic groups: Their associations with sleep quality and impairments	N = 845 (M: 157, F: 688), mean age: 31.66; population: China	A general/health questionnaire;	The significance level was set at 0.05.	Patients without TMD; N = 116 (F: 73, M: 43); mean age: 31.66)	Sleep component scores showed significant differences across severity levels, indicating a worsening trend in sleep quality with increasing TMD severity (*p* < 0.001).
			Fonseca Anamnestic Index (FAI);	Shapiro–Wilks test		Subjects with any DC/TMD diagnoses (PT, IT, or CT) had significantly higher global PSQI scores compared to those with no TMDs (*p* < 0.001).
			Diagnostic Criteria for Temporomandibular disorders (DC/TMD)	Chi-square test		Significant age differences were found between subjects with moderate and severe TMDs (*p* = 0.031) and those with PT, IT, and CT (*p* < 0.001).
			Symptom Questionnaire;	Kruskal–Wallis and Mann–Whitney U post-hoc test		Women were significantly more prevalent than men in all TMD severity and diagnostic groups (*p* < 0.001).
			Pittsburgh sleep quality index (PSQI)	Results were presented as odds ratios (ORs) with 95% confidence intervals (95% CI).		

**Table 4 healthcare-12-00575-t004:** Risk of bias assessed by The Joanna Briggs Institute (JBI) analytical cross-sectional studies’ Critical Appraisal Tool. (Y—Yes, N—No, U—Unclear, L—Low RoB, M—Moderate RoB, H—High RoB).

First Author, Year of Publication/Risk of Bias Assessment Criteria	Criteria for Inclusion in the Sample Are Clearly Defined	Study Subjects and the Setting Described in Detail	Exposure Measured in a Valid and Reliable Way	Objective and Standard Criteria Were Used for the Measurement of the Condition	Confounding Factors Were Identified	Strategies to Deal with Confounding Factors Were Stated	Outcomes Were Measured in a Valid and Reliable Way	Appropriate Statistical Analysis Was Used	Risk of Bias
Cláudia Barbosa, 2021 [21]	Y	Y	Y	Y	Y	N	Y	Y	L
Päivi Jussila, 2018 [28]	N	Y	Y	Y	N	N	Y	Y	H
Bartosz Dalewski, 2021 [29]	Y	Y	Y	Y	Y	Y	Y	Y	L
Louis Simoen, 2020 [23]	Y	Y	Y	Y	Y	N	Y	Y	L
Alessandro Ugolini, 2020 [24]	N	Y	Y	Y	Y	Y	Y	Y	L
Tadej Ostrc, 2022 [22]	N	Y	Y	Y	Y	N	Y	Y	M
Magdalena Osiewicz, 2019 [25]	Y	Y	Y	Y	Y	Y	Y	Y	L
Elisa Tervahauta, 2022 [31]	N	Y	Y	Y	Y	Y	Y	Y	L
Maria Francesca Sfondrini, 2021 [30]	Y	Y	Y	Y	Y	Y	Y	Y	L
Daniela D. S. Rehm, 2019 [27]	U	Y	Y	Y	N	N	Y	Y	H
Adrian Ujin Yap, 2021 [26]	Y	Y	Y	Y	Y	Y	Y	Y	L

**Table 5 healthcare-12-00575-t005:** Analyzed etiologic factors of TMD.

Primary Outcome		Outcome Significance	Trials	No. of Participants (Studies)	Sum of Participants
Biological	Sex	Significant correlation	Cláudia Barbosa, 2021 [21]	1381	3058
			Päivi Jussila, 2018 [28]	1962	
			Louis Simoen, 2020 [23]	243	
			Alessandro Ugolini, 2020 [24]	224	
			Tadej Ostrc, 2022 [22]	109	
			Maria Francesca Sfondrini, 2021 [30]	100	
			Adrian Ujin Yap, 2021 [26]	961	
		Insignificant correlation	Bartosz Dalewski, 2021 [29]	124	233
			Magdalena Osiewicz, 2019 [25]	109	
	Age	Significant correlation	Bartosz Dalewski, 2021 [29]	124	1085
			Adrian Ujin Yap, 2021 [26]	961	
		Insignificant correlation	Cláudia Barbosa, 2021 [21]	1381	1923
			Alessandro Ugolini, 2020 [24]	224	
			Tadej Ostrc, 2022 [22]	109	
			Magdalena Osiewicz, 2019 [25]	109	
			Maria Francesca Sfondrini, 2021 [30]	100	
	Self-reported health condition	Significant correlation	Päivi Jussila, 2018 [28]	1962	1962
	Genetic mutations	Significant correlation	Bartosz Dalewski, 2021 [29]	124	124
	Oral parafunction	Significant correlation	Cláudia Barbosa, 2021 [21]	1381	1490
			Tadej Ostrc, 2022 [22]	109	
		Insignificant correlation	Alessandro Ugolini, 2020 [24]	224	333
			Magdalena Osiewicz, 2019 [25]	109	
	Anatomical features	Significant correlation	Elisa Tervahauta, 2022 [31]	1845	1945
			Maria Francesca Sfondrini, 2021 [30]	100	
		Insignificant correlation	Alessandro Ugolini, 2020 [24]	224	224
	Fibromyalgia	Significant correlation	Päivi Jussila, 2018 [28]	1962	1962
	Migraine, headache	Significant correlation	Päivi Jussila, 2018 [28]	1962	1962
	Gastrointestinal disease	Significant correlation	Päivi Jussila, 2018 [28]	1962	1962
	Rheumatic disease	Significant correlation	Päivi Jussila, 2018 [28]	1962	1962
	Osteoarthritis	Significant correlation	Päivi Jussila, 2018 [28]	1962	1962
	Thyroid disease	Significant correlation	Päivi Jussila, 2018 [28]	1962	1962
Sociological	Education	Insignificant correlation	Tadej Ostrc, 2022 [22]	109	2071
			Päivi Jussila, 2018 [28]	1962	
	Employment status	Significant correlation	Päivi Jussila, 2018 [28]	1962	1962
	Living conditions	Insignificant correlation	Päivi Jussila, 2018 [28]	1962	1962
	Socioeconomic status	Insignificant correlation	Päivi Jussila, 2018 [28]	1962	1962
	Marital Status	Insignificant correlation	Tadej Ostrc, 2022 [22]	109	2071
			Päivi Jussila, 2018 [28]	1962	
Psychological	Somatization	Significant correlation	Alessandro Ugolini, 2020 [24]	224	333
			Tadej Ostrc, 2022 [22]	109	
		Insignificant correlation	Magdalena Osiewicz, 2019 [25]	109	109
	Sleep quality	Significant correlation	Tadej Ostrc, 2022 [22]	109	2713
			Daniela D. S. Rehm, 2019 [27]	1643	
			Adrian Ujin Yap, 2021 [26]	961	
	Anxiety	Significant correlation	Louis Simoen, 2020 [23]	243	576
			Alessandro Ugolini, 2020 [24]	224	
			Tadej Ostrc, 2022 [22]	109	
	Depression	Significant correlation	Päivi Jussila, 2018 [28]	1962	2647
			Louis Simoen, 2020 [23]	243	
			Alessandro Ugolini, 2020 [24]	224	
			Tadej Ostrc, 2022 [22]	109	
			Magdalena Osiewicz, 2019 [25]	109	
	Stress	Insignificant correlation	Magdalena Osiewicz, 2019 [25]	109	109
	Chronic pain	Insignificant correlation	Magdalena Osiewicz, 2019 [25]	109	109

## Data Availability

Data supporting the reported results can be obtained from the first author.

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
