# Peer review of "Etiologic Factors of Temporomandibular Disorders: A Systematic Review of Literature Containing Diagnostic Criteria for Temporomandibular Disorders (DC/TMD) and Research Diagnostic Criteria for Temporomandibular Disorders (RDC/TMD) from 2018 to 2022"

_healthcare, 2024, doi:10.3390/healthcare12050575_

Round 1
Reviewer 1 Report
Comments and Suggestions for Authors
Dear Authors,
thank you for the interesting article. Although I find the quality of this review pretty high, I would suggest some improvements:
1. "Temporomandibular disorders and oral parafunctions: mechanism, diagnostics, and therapy." - this should be the main goal to be presented in the introduction. Please, focus on the multifactorial background of the TMDs and tick some of the systemic diseases connected to that topic: hupertenstion (eg. Martynowicz H, Dymczyk P, Dominiak M, Kazubowska K, Skomro R, Poreba R, Gac P, Wojakowska A, Mazur G, Wieckiewicz M. Evaluation of Intensity of Sleep Bruxism in Arterial Hypertension. J Clin Med. 2018 Oct 5;7(10):327. doi: 10.3390/jcm7100327.), problems with sugar / insuline regulations, SAPHO syndrome, rheumatoid arthritis etc.
2. Also, present the diagnostic intruments, especially if some of them were validated in your country (search for Diagnostic Criteria for Temporomandibular Disorders (DC/TMD)). Also, present them as in finding the relationship between probable sleep bruxism, awake bruxism and temporomandibular disorders.
3. Line 43, after the name of the Author, provide the reference (use it in the whole manuscript) - line 56, 58 etc
4. Delete space in line 60, before the sentence begins
5. Because of the search exclusion criteria (no. of participants <100, case reports), you have to refer to them as the limitations at the end of the paper, which had been partially done by you. That would also be an interesting study to explore it further and begin the metaanalysis (as for future plan.). Also, the data from the countries are limited to mostly Europe (but also China and Brazil)
6. What exactly do you mean by "study design" (prisma flow diagram) - refer to it in the main text
7. Give the paragraph 3.4 more descriptive pattetn. The ones presented now should be either a graph or presented as table
8. Line 368 - what do you mean by "older age"
9. I also think you missed one of the recently published papers:
Seweryn P, Orzeszek SM, Waliszewska-Prosół M, Jenča A, Osiewicz M, Paradowska-Stolarz A, Winocur-Arias O, Ziętek M, Bombała W, Więckiewicz M. Relationship between pain severity, satisfaction with life and the quality of sleep in Polish adults with temporomandibular disorders. Dent Med Probl. 2023 Oct-Dec;60(4):609-617. doi: 10.17219/dmp/171894
that sums up the life satistaction / mental problems caused by TMD, which relates strongly to paragraphs 4.2, 4.5, 4.6, 4.8, 4.9
Beside those small flaws and small suggestions, the paper is well prepared and definatelly deserves publishing.
Author Response
Dear Reviewer,
We would like to express our sincere gratitude for your thorough review and constructive feedback. We have carefully considered your suggestions and made the following revisions to the manuscript:
1. Introduction:
- The section on etiology has been expanded to include the suggested factors, as well as a few additional ones. This was a very valuable suggestion, and we thank you for pointing out that this aspect was missing from the introduction.
2. Reference to TMD diagnostic tools has been added.
3. & 4. Corrected as suggested.
5. Suggested limitations have been added to the "Limitations of the Study" section.
6. Studies rejected due to "study design" are those that were rejected due to difficulty in interpretation (unclear results). These were not described in the main text because we considered this to be a relatively minor aspect.
7. We consciously chose this format, as we believe it is more readable.
8. "Older age" was changed to "age" due to the differences between the studies and the lack of guidelines on this issue.
9. This study was not included in our review because it was published after the literature review was completed.
We greatly appreciate your insightful and detailed analysis of our work. Your feedback has allowed us to improve the quality of the manuscript, which is very important to us.
Sincerely,
The Authors
Reviewer 2 Report
Comments and Suggestions for Authors
Thank you for conducting this interesting topic. I have the following comments:
· Why the search period was short?
· The data base of searching is few
· The followings were not searched: the reference lists / bibliographies of included studies, trial/study registries
· The authors did not provide a list of excluded studies and justify the exclusions
· The two studies with high risk of bias should be excluded from the discussion to avoid misleading outcomes from high-risk studies
· The review authors did not report on the sources of funding for the studies included in the review
· The review authors did not account for RoB in individual studies when interpreting/ discussing the results of the review
The above comments reduced the level of evidence for the systematic review
Author Response
Dear Reviewer,
Thank you very much for your analysis of the paper and your comments.
1 The study period was 5 years because we were concerned with the latest research based on DC/TMD, which has only recently begun to be more widely used.
2. This fact was described, in the limitation of the work.
3. 1467 papers were excluded, which makes it impossible to provide a complete list of excluded papers. The reasons for the exclusion of papers are included in the prisma flow diagram.
4.We do have performed an extensive review of the literature included in this review. The papers we selected appeared to be relevant enough to be included in the analysis despite the high risk of RoB.
5.The paper was written in the free time of the authors, who are not academics; therefore, there was no external source of funding.
6. Has been corrected.
We appreciate your analysis and hope that the corrections made will improve the quality of our work.
Reviewer 3 Report
Comments and Suggestions for Authors Dear authors, thank you for the work presented. I leave the document with some comments and suggestions, for your consideration and alterations in the manuscript.Kind regards.

Author Response
Dear Reviewer,
thank you for your comments. We have made corrections to your comments, but some of the remarks were not clear to us, so if there is anything else to improve, please let us know.
At the same time, thank you very much for your comment about bruxism-you are right, we overlooked it.
We have checked paperwork 14 and 15 again, and to our knowledge, the authors also used Axis II. The limitations of the work have also been corrected.
Sincerely,
Authors
Round 2
Reviewer 1 Report
Comments and Suggestions for Authors
Dear Authors,
thank you for providing all corrections. The paper could be accepted in this way.
The repetitions refer mainly to names of the institutions and the methodology (that is written in this pattern always), so I think it is acceptable.
Thank you